# A Simple Reward-free Approach to Constrained Reinforcement Learning

## Abstract

In constrained reinforcement learning (RL), a learning agent seeks to not only optimize the overall reward but also satisfy the additional safety, diversity, or budget constraints. Consequently, existing constrained RL solutions require several new algorithmic ingredients that are notably different from standard RL. On the other hand, reward-free RL is independently developed in the unconstrained literature, which learns the transition dynamics without using the reward information, and thus naturally capable of addressing RL with multiple objectives under the common dynamics. This paper bridges reward-free RL and constrained RL. Particularly, we propose a simple meta-algorithm such that given any reward-free RL oracle, the approachability and constrained RL problems can be directly solved with negligible overheads in sample complexity. Utilizing the existing reward-free RL solvers, our framework provides sharp sample complexity results for constrained RL in the tabular MDP setting, matching the best existing results up to a factor of horizon dependence; our framework directly extends to a setting of tabular two-player Markov games, and gives a new result for constrained RL with linear function approximation.

## 1 Introduction

In a wide range of modern reinforcement learning (RL) applications, it is not sufficient for the learning agents to only maximize a scalar reward. More importantly, they must satisfy various *constraints*. For instance, such constraints can be the physical limit of power consumption or torque in motors for robotics tasks (Tessler et al., 2019); the budget for computation and the frequency of actions for real-time strategy games (Vinyals et al., 2019); and the requirement for safety, fuel efficiency and human comfort for autonomous drive (Le et al., 2019). In addition, constraints are also crucial in tasks such as dynamic pricing with limited supply (Besbes & Zeevi, 2009; Babaioff et al., 2015), scheduling of resources on a computer cluster (Mao et al., 2016), imitation learning (Syed & Schapire, 2007; Ziebart et al., 2008; Sun et al., 2019), as well as RL with fairness (Jabbari et al., 2017).

These huge demand in practice gives rise to a subfield—constrained RL, which focuses on designing efficient algorithms to find near-optimal policies for RL problems under linear or general convex constraints. Most constrained RL works directly combine the existing techniques such as value iteration and optimism from unconstrained literature, with new techniques specifically designed to deal with linear constraints (Efroni et al., 2020; Ding et al., 2021; Qiu et al., 2020) or general convex constraints (Brantley et al., 2020; Yu et al., 2021). The end product is a single new complex algorithm which is tasked to solve all the challenges of learning dynamics, exploration, planning as well as constraints satisfaction simultaneously. Thus, these algorithms need to be re-analyzed from scratch, and it is highly nontrivial to translate the progress in the unconstrained RL to the constrained setting.

On the other hand, reward-free RL—proposed in Jin et al. (2020a)—is a framework for the unconstrained setting, which learns the transition dynamics without using the reward. The framework has two phases: in the exploration phase, the agent first collects trajectories from a Markov decision process (MDP) and learns the dynamics without a pre-specified reward function. After exploration, the agent is tasked with computing near-optimal policies under the MDP for a collection of given reward functions. This framework is particularly suitable when there are multiple reward functions of interest, and has been developed recently to attack various settings including tabular MDPs (Jin et al.,

Table 1: Sample complexity for algorithms to solve reward-free RL for VMDP (Definition 1), approachability (Definition 3) and CMDP with general convex constraints (Definition 4).[1]

| | Algorithm | Reward-free | Approachability | CMDP |
|---|---|---|---|---|
| Tabular | Wu et al. (2020) | $\tilde{\mathcal{O}}(\min\{d, S\}H^4SA/\epsilon^2)$ | - | - |
| | Brantley et al. (2020) | - | - | $\tilde{\mathcal{O}}(d^2H^3S^2A/\epsilon^2)$ |
| | Yu et al. (2021) | - | $\tilde{\mathcal{O}}(\min\{d, S\}H^3SA/\epsilon^2)$ | $\tilde{\mathcal{O}}(\min\{d, S\}H^3SA/\epsilon^2)$ |
| | This work | $\tilde{\mathcal{O}}(\min\{d, S\}H^4SA/\epsilon^2)$ | $\tilde{\mathcal{O}}(\min\{d, S\}H^4SA/\epsilon^2)$ | $\tilde{\mathcal{O}}(\min\{d, S\}H^4SA/\epsilon^2)$ |
| Linear | This work | $\tilde{\mathcal{O}}(d_{\text{lin}}^3 H^6/\epsilon^2)$ | $\tilde{\mathcal{O}}(d_{\text{lin}}^3 H^6/\epsilon^2)$ | $\tilde{\mathcal{O}}(d_{\text{lin}}^3 H^6/\epsilon^2)$ |

2020a; Zhang et al., 2020), linear MDPs (Wang et al., 2020; Zanette et al., 2020), and tabular Markov games (Liu et al., 2020).

**Contribution.** In this paper, we propose a simple approach to solve constrained RL problems by bridging the reward-free RL literature and constrained RL literature. Our approach isolates the challenges of constraint satisfaction, and leaves the remaining RL challenges such as learning dynamics and exploration to reward-free RL. This allows us to design a new algorithm which purely focuses on addressing the constraints. Formally, we design a meta-algorithm for RL problems with general convex constraints. Our meta-algorithm takes a reward-free RL solver, and can be used to directly solve the approachability problem, as well as the constrained MDP problems using very small amount of samples in addition to what is required for reward-free RL.

Our framework enables direct translation of any progress in reward-free RL to constrained RL. Leveraging recent advances in reward-free RL, our meta-algorithm directly implies sample-efficient guarantees of constrained RL in the settings of tabular MDP, linear MDP, as well as tabular two-player Markov games. In particular,

- *Tabular setting*: Our work achieves sample complexity of $\tilde{\mathcal{O}}(\min\{d, S\}H^4SA/\epsilon^2)$ for all three tasks of reward-free RL for Vector-valued MDPs (VMDP), approachability, and RL with general convex constraints. Here $d$ is the dimension of VMDP or the number of constraints, $S, A$ are the number of states and actions, $H$ is the horizon, and $\epsilon$ is the error tolerance. It matches the best existing results up to a factor of $H$.

- *Linear setting*: Our work provides new sample complexity of $\tilde{\mathcal{O}}(d_{\text{lin}}^3 H^6/\epsilon^2)$ for all three tasks above for linear MDPs. To our best knowledge, this result is the first sample-efficient result for approachability and also constrained RL with general convex constraints in the linear function approximation setting.

- *Two-player setting*: Our work extends to the setting of tabular two-player vector-valued Markov games and achieves low regret of $\alpha(T) = \mathcal{O}(\epsilon/2 + \sqrt{H^2\iota/T})$ at the cost of this $\mathcal{O}(\epsilon)$ bias in regret as well as additional samples for preprocessing.

### 1.1 RELATED WORK

In this section, we review the related works on three tasks studied in this paper—reward-free RL, approachability, and constrained RL.

**Reward-free RL.** Reward-free exploration has been formalized by Jin et al. (2020a) for the tabular setting. Furthermore, Jin et al. (2020a) proposed an algorithm which has sample complexity $\tilde{\mathcal{O}}(\text{poly}(H)S^2A/\epsilon^2)$ outputting $\epsilon$-optimal policy for arbitrary number of reward functions. More

---

[1]The presented sample complexities are all under the $L_2$ normalization conditions as studied in this paper. We comment that the results of (Wu et al., 2020; Brantley et al., 2020; Yu et al., 2021) are originally presented under $L_1/L_\infty$ normalization conditions. While the results in Wu et al. (2020) can be directly adapted to our setting as stated in the table, the other two results Brantley et al. (2020); Yu et al. (2021) will be no better than the displayed results after adaptation.

recently, Zhang et al. (2020); Liu et al. (2020) propose algorithm VI-Zero with sharp sample complexity of $\tilde{\mathcal{O}}(\text{poly}(H)\log(N)SA/\epsilon^2)$ capable of handling $N$ fixed reward functions. Wang et al. (2020); Zanette et al. (2020) further provide reward-free learning results in the setting of linear function approximation, in particular, Wang et al. (2020) guarantees to find the near-optimal policies for an arbitrary number of (linear) reward functions within a sample complexity of $\tilde{\mathcal{O}}(\text{poly}(H)d_{\text{lin}}^3/\epsilon^2)$. All results mentioned above are for scalar-valued MDPs. For the vector-valued MDPs (VMDPs), very recent work of Wu et al. (2020) designs a reward-free algorithm with sample complexity guarantee $\tilde{\mathcal{O}}(\text{poly}(H)\min\{d,S\}SA/\epsilon^2)$ in the tabular setting. Compared to Wu et al. (2020), our reward-free algorithms for VMDP is adapted from the VI-Zero algorithm presented in Liu et al. (2020); While achieving the same sample complexity, it allows arbitrary planning algorithms in the planning phase.

**Approachability and Constrained RL**  Approachability and Constrained RL are two related tasks involving constraints. Inspired by Blackwell approachability (Blackwell et al., 1956), recent work of Miryoosefi et al. (2019) introduces approachability task for VMDPs. However, the proposed algorithm does not have polynomial sample complexity guarantees. More recently, Yu et al. (2021) gave a new algorithm for approachability for both VMDPs and vector-valued Markov games (VMGs). Yu et al. (2021) provides regret bounds for the proposed algorithm resulting in sample complexity guarantees of $\tilde{\mathcal{O}}(\text{poly}(H)\min\{d,S\}SA/\epsilon^2)$ for approachability in VMDPs and $\tilde{\mathcal{O}}(\text{poly}(H)\min\{d,S\}SAB/\epsilon^2)$ for approachability in VMGs.

Sample-efficient exploration in constrained reinforcement learning has been recently studied in a recent line of work by Brantley et al. (2020); Qiu et al. (2020); Efroni et al. (2020); Ding et al. (2021); Singh et al. (2020). All these works are also limited to linear constraints except Brantley et al. (2020) which extends their approach to general convex constraints achieving sample complexity of $\tilde{\mathcal{O}}(\text{poly}(H)d^2S^2A/\epsilon^2)$ . However, Brantley et al. (2020) requires solving a large-scale convex optimization sub-problem. The best result for constrained RL with general convex constraints can be achieved by the approachability-based algorithm in Yu et al. (2021) obtaining sample complexity of $\tilde{\mathcal{O}}(\text{poly}(H)\min\{d,S\}SA/\epsilon^2)$. Technically, our meta-algorithm is based on the Fenchel's duality, which is similar to Yu et al. (2021). In contrast, Yu et al. (2021) does not use reward-free RL, and is thus different from our results in terms of algorithmic approaches. Consequently, Yu et al. (2021) does not reveal the deep connections between reward-free RL and constrained RL, which is one of the main contribution of this paper. In addition, Yu et al. (2021) does not address the function approximation setting.

Finally, we note that among all results mentioned above, only Ding et al. (2021) has considered models beyond tabular setting in the context of constrained RL. The model studied in Ding et al. (2021) is known as linear mixture MDPs which is different and incomparable to the linear MDP models considered in this paper. We further comment that Ding et al. (2021) can only handle linear constraints for CMDP, while our results is capable of solving CMDPs with general convex constraints.

## 2  PRELIMINARIES AND PROBLEM SETUP

We consider an episodic *vector-valued Markov decision process* (VMDP) specified by a tuple $\mathcal{M} = (\mathcal{S}, \mathcal{A}, H, \mathbb{P}, \mathbf{r})$, where $\mathcal{S}$ is the state space, $\mathcal{A}$ is the action space, $H$ is the length of each episode, $\mathbb{P} = \{\mathbb{P}_h\}_{h=1}^H$ is the collection of *unknown* transition probabilities with $\mathbb{P}_h(s' \mid s, a)$ equal to the probability of transiting to $s'$ after taking action $a$ in state $s$ at the $h^{\text{th}}$ step, and $\mathbf{r} = \{\mathbf{r}_h : \mathcal{S} \times \mathcal{A} \to \mathcal{B}(1)\}_{h=1}^H$ is a collection of *unknown* $d$-dimensional return functions, where $\mathcal{B}(r)$ is the $d$-dimensional Euclidean ball of radius $r$ centered at the origin.

**Interaction protocol.**  In each episode, agent starts at a *fixed* initial state $s_1$. Then, at each step $h \in [H]$, the agent observes the current state $s_h$, takes action $a_h$, receives stochastic sample of the return vector $\mathbf{r}_h(s_h, a_h)$, and it causes the environment to transit to $s_{h+1} \sim \mathbb{P}_h(\cdot \mid s_h, a_h)$. We assume that stochastic samples of the return function are also in $\mathcal{B}(1)$, almost surely.

**Policy and value function.**  A policy $\pi$ of an agent is a collection of $H$ functions $\{\pi_h : \mathcal{S} \to \Delta(\mathcal{A})\}_{h=1}^H$ that map states to distribution over actions. The agent following policy $\pi$, picks action $a_h \sim \pi_h(s_h)$ at the $h^{\text{th}}$ step. We denote $\mathbf{V}_h^\pi : \mathcal{S} \to \mathcal{B}(H)$ as the value func-

tion at step $h$ for policy $\pi$, defined as $\mathbf{V}_h^\pi(s) := \mathbb{E}_\pi \left[ \sum_{h'=h}^H \mathbf{r}_{h'}(s_{h'}, a_{h'}) \mid s_h = s \right]$. Similarly, we denote $\mathbf{Q}_h^\pi : \mathcal{S} \times \mathcal{A} \to \mathcal{B}(H)$ as the $Q$-value function at step $h$ for policy $\pi$, where $\mathbf{Q}_h^\pi(s, a) := \mathbb{E}_\pi \left[ \sum_{h'=h}^H \mathbf{r}_{h'}(s_{h'}, a_{h'}) \mid s_h = s, a_h = a \right]$.

**Scalarized MDP.** For a VMDP $\mathcal{M}$ and $\boldsymbol{\theta} \in \mathcal{B}(1)$, we define scalar-valued MDP $\mathcal{M}_{\boldsymbol{\theta}} = (\mathcal{S}, \mathcal{A}, H, \mathbb{P}, r_{\boldsymbol{\theta}})$, where $r_{\boldsymbol{\theta}} = \{\langle \boldsymbol{\theta}, \mathbf{r}_h \rangle : \mathcal{S} \times \mathcal{A} \to [-1, 1]\}_{h=1}^H$. We denote $V_h^\pi(\cdot; \boldsymbol{\theta}) : \mathcal{S} \to [-H, H]$ as the scalarized value function at step $h$ for policy $\pi$, defined as

$$V_h^\pi(s; \boldsymbol{\theta}) := \mathbb{E}_\pi \left[ \sum_{h'=h}^H \langle \boldsymbol{\theta}, \mathbf{r}_{h'}(s_{h'}, a_{h'}) \rangle \mid s_h = s \right] = \langle \boldsymbol{\theta}, \mathbf{V}_h^\pi(s) \rangle.$$

Similarly, we denote $Q_h^\pi(\cdot; \boldsymbol{\theta}) : \mathcal{S} \times \mathcal{A} \to [-H, H]$ as the scalarized $Q$-value function at step $h$ for policy $\pi$, where

$$Q_h^\pi(s, a; \boldsymbol{\theta}) := \mathbb{E}_\pi \left[ \sum_{h'=h}^H \langle \boldsymbol{\theta}, \mathbf{r}_{h'}(s_{h'}, a_{h'}) \rangle \mid s_h = s, a_h = a \right] = \langle \boldsymbol{\theta}, \mathbf{Q}_h^\pi(s, a) \rangle.$$

For a fixed $\boldsymbol{\theta} \in \mathbb{R}^d$, there exists an optimal policy $\pi_{\boldsymbol{\theta}}^\star$, maximizing value for all states (Puterman, 2014); i.e., $V_h^{\pi_{\boldsymbol{\theta}}^\star}(s; \boldsymbol{\theta}) = \sup_\pi V_h^\pi(s; \boldsymbol{\theta})$ for all $s \in \mathcal{S}$ and $h \in [H]$. We abbreviate $V^{\pi_{\boldsymbol{\theta}}^\star}(\cdot; \boldsymbol{\theta})$ and $Q^{\pi_{\boldsymbol{\theta}}^\star}(\cdot; \boldsymbol{\theta})$ as $V^\star(\cdot; \boldsymbol{\theta})$ and $Q^\star(\cdot; \boldsymbol{\theta})$ respectively.

## 2.1 REWARD-FREE EXPLORATION (RFE) FOR VMDPs

The task of *reward-free exploration* (formalized by Jin et al. (2020a) for tabular MDPs) considers the scenario in which the agents interacts with the environment without guidance of reward information. Later, the reward information is revealed and the agents is required to compute the near-optimal policy. In this section, we describe its counterpart for VMDPs [1]. Formally, it consists of two phases:

**Exploration phase.** In the exploration phase, agent explores the unknown environment without observing any information regarding the return function. Namely, at each episode the agent executes policies to collect samples. The policies can depend on dynamic observations $\{s_h^k, a_h^k\}_{(k,h) \in [K] \times [H]}$ in the past episodes, but not the return vectors.

**Planning phase.** In the planning phase, the agent no longer interacts with the environment; however, stochastic samples of the $d$-dimensional return function for the collected episodes is revealed to the agent, i.e. $\{\mathbf{r}_h^k\}_{(k,h) \in [K] \times [H]}$. Based on the episodes collected during the exploration phase, the agent outputs the near-optimal policies of $\mathcal{M}_{\boldsymbol{\theta}}$ given an arbitrary number of vectors $\boldsymbol{\theta} \in \mathcal{B}(1)$.

**Definition 1** (Reward-free algorithm for VMDPs). *For any $\epsilon, \delta > 0$, after collecting $m_{\mathrm{RFE}}(\epsilon, \delta)$ episodes during the exploration phase, with probability at least $1 - \delta$, the algorithm satisfies*

$$\forall \boldsymbol{\theta} \in \mathcal{B}(1) : \quad V_1^\star(s_1; \boldsymbol{\theta}) - V_1^{\pi_{\boldsymbol{\theta}}}(s_1; \boldsymbol{\theta}) \le \epsilon, \tag{1}$$

*where $\pi_{\boldsymbol{\theta}}$ is the output of the planning phase for vector $\boldsymbol{\theta}$ as input. The function $m_{\mathrm{RFE}}$ determines the sample complexity of the RFE algorithm.*

**Remark 2.** *Standard reward-free setup concerns MDPs with scalar reward, and requires the algorithm to find the near-optimal policies for $N$ different prespecified reward functions in the planning phase, where the sample complexity typically scales with $\log N$. This type of results can be adapted into a guarantee in the form of (1) for VMDP by $\epsilon$-covering of $\boldsymbol{\theta}$ over $\mathcal{B}(1)$ and a modified concentration arguments (see the proofs of Theorem 7 and Theorem 13 for more details).*

## 2.2 APPROACHABILITY FOR VMDPs

In this section we provide the description for the *approachability* task for VMPDs introduced by Miryoosefi et al. (2019). Given a vector-valued Markov decision process and a convex target set $\mathcal{C}$, the goal is to learn a policy whose expected cumulative return vector lies in the target set (akin to Blackwell approachability in single-turn games, Blackwell et al. 1956). We consider the agnostic version of this task which is more general since it doesn't need to assume that such policy exists; instead, the agent learns to minimize the Euclidean distance between expected return of the learned policy and the target set.

---

[1]RFE for VMDPs is also called preference-free exploration problem in Wu et al. (2020)

**Definition 3** (Approachability algorithm for VMDPs). *For any $\epsilon, \delta > 0$, after collecting $m_{\text{APP}}(\epsilon, \delta)$ episodes, with probability at least $1 - \delta$, the algorithm satisfies*

$$\text{dist}(\mathbf{V}_1^{\pi^{\text{out}}}(s_1), \mathcal{C}) \leq \min_\pi \text{dist}(\mathbf{V}_1^\pi(s_1), \mathcal{C}) + \epsilon, \tag{2}$$

*where $\pi^{\text{out}}$ is the output of the algorithm and $\text{dist}(\mathbf{x}, \mathcal{C})$ is the Euclidean distance between point $\mathbf{x}$ and set $\mathcal{C}$. The function $m_{\text{APP}}$ determines the sample complexity of the algorithm.*

### 2.3 Constrained MDP (CMDP) with general convex constraints

In this section we describe *constrained Markov decision processes* (CMDPs) introduced by Altman (1999). The goal of this setting is to minimize cost while satisfying some linear constraints over consumption of $d$ resources (resources are akin to $\mathbf{r}$ in our case). Although, the original definition only allows for linear constraints, we consider the more general case of arbitrary convex constraints. More formally, consider a VMDP $\mathcal{M}$, a cost function $c = \{c_h : \mathcal{S} \times \mathcal{A} \to [-1, 1]\}_{h=1}^H$, and a convex constraint set $\mathcal{C}$. The agent goal is to compete against the following benchmark:

$$\min_\pi C_1^\pi(s_1) \quad \text{s.t.} \quad \mathbf{V}_1^\pi(s_1) \in \mathcal{C},$$

where $C_h^\pi = \mathbb{E}_\pi \left[ \sum_{h'=h}^H c_{h'}(s_{h'}, a_{h'}) \mid s_h = s \right]$.

**Definition 4** (Algorithm for CMDP). *For any $\epsilon, \delta > 0$, after collecting $m_{\text{CMDP}}(\epsilon, \delta)$ episodes, with probability at least $1 - \delta$, the algorithm satisfies*

$$\begin{cases} C_1^{\pi^{\text{out}}}(s_1) - \min_{\pi : \mathbf{V}_1^\pi(s_1) \in \mathcal{C}} C_1^\pi(s_1) \leq \epsilon \\ \text{dist}\left(\mathbf{V}_1^{\pi^{\text{out}}}(s_1), \mathcal{C}\right) \leq \epsilon, \end{cases} \tag{3}$$

*where $\pi^{\text{out}}$ is the output of the algorithm. The function $m_{\text{CMDP}}$ determines the sample complexity of the algorithm.*

As also mentioned in the prior works (Miryoosefi et al., 2019; Yu et al., 2021), we formally show in the following theorem that approachability task (Definition 3) can be considered more general compared to CMDP (Definition 4); Namely, given any algorithm for the former we can obtain an algorithm for the latter by incurring only extra logarithmic factor and a negligible overhead. The idea is to incorporate cost into the constraint set $\mathcal{C}$ and perform an (approximate) binary search over the minimum attainable cost. The reduction and the proof can be found in Appendix A.

**Theorem 5.** *Given any approachability algorithm (Definition 3) with sample complexity $m_{\text{APP}}$, we can design an algorithm for CMDP (Definition 4) with sample complexity $m_{\text{CMDP}}$, satisfying*

$$m_{\text{CMDP}}(\epsilon, \delta) \leq \mathcal{O}\left( m_{\text{APP}}\left(\frac{\epsilon}{6}, \frac{\epsilon\delta}{12H}\right) + \frac{H^2 \log[dH/\epsilon\delta]}{\epsilon^2} \right) \cdot \log \frac{1}{\epsilon}.$$

## 3 Meta-algorithm for VMDPs

In this section, equipped with preliminaries discussed in Section 2, we are ready introduce our main algorithmic framework for VMDPs bridging reward-free RL and approachability.

Before introducing the algorithm, we explain the intuition behind it. By Fenchel's duality (similar to Yu et al. 2021), one can show that

$$\min_\pi \text{dist}(\mathbf{V}_1^\pi(s_1, \mathcal{C})) = \min_\pi \max_{\theta \in \mathcal{B}(1)} \left[ \langle \boldsymbol{\theta}, \mathbf{V}_1^\pi(s_1, \mathcal{C}) \rangle - \max_{\mathbf{x}' \in \mathcal{C}} \langle \boldsymbol{\theta}, \mathbf{x}' \rangle \right].$$

It satisfies the minimax conditions since it's concave in $\boldsymbol{\theta}$ and convex in $\pi$ (by allowing mixture policies); therefore, minimax theorem Neumann (1928) implies that we can equivalently solve

$$\max_{\theta \in \mathcal{B}(1)} \min_\pi \left[ \langle \boldsymbol{\theta}, \mathbf{V}_1^\pi(s_1, \mathcal{C}) \rangle - \max_{\mathbf{x}' \in \mathcal{C}} \langle \boldsymbol{\theta}, \mathbf{x}' \rangle \right].$$

This max-min form allows us to use general technique of Freund & Schapire (1999) for solving a max-min by repeatedly playing a no-regret online learning algorithm as the max-player against best-response for the min-player. In particular, for a fixed $\boldsymbol{\theta}$, minimizing over $\pi$ is equivalent to finding optimal policy for scalarized MDP $\mathcal{M}_{-\boldsymbol{\theta}}$. To achieve this, we can utilize a reward-free oracle as in Definition 1. On the other hand for $\boldsymbol{\theta}$-player we are able to use online gradient descent (Zinkevich, 2003). By combining ideas above, we obtain Algorithm 1.

---

**Algorithm 1** Meta-algorithm for VMDPs

---

1: **Input:** Reward-Free Algorithm RFE for VMDPs (as in Definiton 1), Target Set $\mathcal{C}$
2: **Hyperparameters:** learning rate $\eta^t$
3: **Initialize:** run exploration phase of RFE for $K$ episodes
4: **Set:** $\boldsymbol{\theta}^1 \in \mathcal{B}(1)$
5: **for** $t = 1, 2, \ldots, T$ **do**
6:     Obtain near optimal policy for $\mathcal{M}_{-\boldsymbol{\theta}^t}$:

$$\pi^t \leftarrow \text{output of planning phase of RFE for preference vector } -\boldsymbol{\theta}^t$$

7:     Estimate $\mathbf{V}_1^{\pi^t}(s_1)$ using one episode:

$$\text{Run } \pi^t \text{ for one episode and let } \widehat{\mathbf{v}}^t \text{ be the sum of vectorial returns}$$

8:     Apply online gradient ascent update for utility function $u^t(\boldsymbol{\theta}) = \langle \boldsymbol{\theta}, \widehat{\mathbf{v}}^t \rangle - \max_{\mathbf{x} \in \mathcal{C}} \langle \boldsymbol{\theta}, \mathbf{x} \rangle$:

$$\boldsymbol{\theta}^{t+1} \leftarrow \Gamma_{\mathcal{B}(1)}[\boldsymbol{\theta}^t + \eta^t(\widehat{\mathbf{v}}^t - \text{argmax}_{\mathbf{x} \in \mathcal{C}} \langle \boldsymbol{\theta}^t, \mathbf{x} \rangle)]$$

    where $\Gamma_{\mathcal{B}(1)}$ is the projection into Euclidean unit ball
9: Let $\pi^{\text{out}}$ be uniform mixture of $\{\pi^1, \ldots, \pi^T\}$
10: **Return** $\pi^{\text{out}}$

---

**Theorem 6.** *There exists an absolute constant c, such that for any choice of RFE algorithm (Definition 1) and for any $\epsilon \in (0, H]$ and $\delta \in (0, 1]$, if we choose*

$$T \geq c(H^2 \iota / \epsilon^2), \quad K \geq m_{\text{RFE}}(\epsilon/2, \delta/2), \text{ and } \quad \eta^t = \sqrt{1/(H^2 t)},$$

*where $\iota = \log(d/\delta)$; then, with probability at least $1 - \delta$, Algorithm 1 outputs an $\epsilon$-optimal policy for the approachability (Equation 2). Therefore, we have $m_{\text{APP}}(\epsilon, \delta) \leq \mathcal{O}(m_{\text{RFE}}(\epsilon/2, \delta/2) + H^2 \iota / \epsilon^2)$.*

Theorem 6 shows that given any reward-free algorithm, Algorithm 1 can solve the approachability task with negligible overhead. The proof for Theorem 6 is provided in Appendix B. Equipped with this theorem, since we have already shown the connection between approachability and constrained RL in Theorem 5, any results for RFE can be directly translated to results for constrained RL.

## 4 TABULAR VMDPS

In this section, we consider tabular VMDPs; namely, we assume that $|\mathcal{S}| \leq S$ and $|\mathcal{A}| \leq A$. Utilizing prior work on tabular setting, we describe our choice of reward-free algorithm.

In the exploration phase, we use VI-Zero proposed by Liu et al. (2020). It can be seen as UCB-VI (Azar et al., 2017) with zero reward. Intuitively, the value function computed in the algorithm measures the level of uncertainty and incentivizes the greedy policy to visit underexplored states. The output of VI-Zero is $\widehat{\mathbb{P}}^{\text{out}}$, which is an estimation of the transition dynamics.

In the planning phase, given $\boldsymbol{\theta} \in \mathcal{B}(1)$ we can use any planning algorithm (e.g., value iteration) for $\widehat{\mathcal{M}}_{\boldsymbol{\theta}} = (\mathcal{S}, \mathcal{A}, H, \widehat{\mathbb{P}}^{\text{out}}, \langle \boldsymbol{\theta}, \widehat{\mathbf{r}} \rangle)$ where $\widehat{\mathbf{r}}$ is empirical estimate of $\mathbf{r}$ using collected samples $\{\mathbf{r}_h^k\}$.

The following theorem state theoretical guarantees for tabular VMDPs. Proof of Theorem 7 and more details can be found in Appendix C.

**Theorem 7.** *For tabular VMDP, we have a reward-free algorithm (Definiton 1) with $m_{\text{RFE}}(\epsilon, \delta) \leq \mathcal{O}(\min\{d, S\}H^4 SA\iota/\epsilon^2 + H^3 S^2 A\iota^2/\epsilon)$, an algorithm for approachability (Definition 3) with $m_{\text{APP}}(\epsilon, \delta) \leq \mathcal{O}(\min\{d, S\}H^4 SA\iota/\epsilon^2 + H^3 S^2 A\iota^2/\epsilon)$, and an algorithm for CMDP (Definition 4) with $m_{\text{CMDP}}(\epsilon, \delta) \leq \mathcal{O}(\min\{d, S\}H^4 SA\iota^2/\epsilon^2 + H^3 S^2 A\iota^3/\epsilon)$.*

The reward-free algorithm with stated sample complexity in Theorem 7 is the VI-Zero algorithm (Algorithm 5 in Appendix C). Its sample complexity result is obtained by adapting the results in Liu et al. (2020) for scalar-valued MDPs to the settings of VMDPs in this paper. The algorithms for approachability and CMDP is based on pluging in VI-Zero into our meta algorithms, and the corresponding sample complexity results are obtained by applying Theorem 5 and our main result— Theorem 6.

Theorem 7 shows that the sample complexity of all three tasks are connected—the leading terms are all $\tilde{\mathcal{O}}(\min\{d, S\}H^4 SA/\epsilon^2)$ which differ by only logarithmic factors. In particular, our sample complexity for the reward-free exploration (Definition 1) in the tabular setting matches the best result in Wu et al. (2020). It further shows that we can easily design an sample-efficient for approachability (Definition 3) and CMDP with general convex constraints (Definition 4) in the tabular setting, with sample complexity matching the best result in Yu et al. (2021) up to a single factor of $H$. [2] Therefore, our framework while being modular enabling direct translation of reward-free RL to constrained RL, achieves sharp sample complexity guarantees. We comment that due to reward-free nature of our approach unlike Yu et al. (2021), we can no longer provide regret guarantees.

# 5 LINEAR FUNCTION APPROXIMATION: LINEAR VMDPs

In this section we consider the setting of linear function approximation and allow $\mathcal{S}$ and $\mathcal{A}$ to be infinitely large. We assume that agent has access to a feature map $\phi : \mathcal{S} \times \mathcal{A} \to \mathbb{R}^{d_{\text{lin}}}$ and the return function and transitions are linear functions of the feature map. We formally define the linear VMDPs in Assumption 8 which adapts the definition of linear MDPs (Jin et al., 2020b) for VMDPS; namely, they coincide for the case of $d = 1$.

**Assumption 8** (Linear VMDP). *A VMDP $\mathcal{M} = (\mathcal{S}, \mathcal{A}, H, \mathbb{P}, \mathbf{r})$ is said to be a linear VMDP with a feature map $\phi : \mathcal{S} \times \mathcal{A} \to \mathbb{R}^{d_{\text{lin}}}$, if for any $h \in [H]$:*

1. *There exists $d_{\text{lin}}$ unknown (signed) measures $\boldsymbol{\mu}_h = \{\mu_h^{(1)}, \ldots, \mu_h^{(d_{\text{lin}})}\}$ over $\mathcal{S}$ such that for any $(s, a) \in \mathcal{S} \times \mathcal{A}$ we have $\mathbb{P}_h(\cdot \mid s, a) = \langle \boldsymbol{\mu}(\cdot), \phi(s, a) \rangle$.*

2. *There exists an unknown matrix $W_h \in \mathbb{R}^{d \times d_{\text{lin}}}$ such that for any $(s, a) \in \mathcal{S} \times \mathcal{A}$ we have $\mathbf{r}_h(s, a) = W_h \phi(s, a)$.*

Similar to Jin et al. (2020b), we assume that $\|\phi(s, a)\| \leq 1$ for all $(s, a) \in \mathcal{S} \times \mathcal{A}$, $\|\boldsymbol{\mu}_h(\mathcal{S})\| \leq \sqrt{d_{\text{lin}}}$ for all $h \in [H]$, and $\|W_h\| \leq \sqrt{d_{\text{lin}}}$ for all $h \in [H]$.

Wang et al. (2020) has recently proposed a sample-efficient algorithm for reward-free exploration in linear MDPs. Utilizing that algorithm and tailoring it for our setting, we can obtain the following theoretical guarantee. The algorithm and the proof can be found in Appendix D.

**Theorem 9.** *For linear VMDPs (Assumption 8), we have a reward-free algorithm (Definiton 1) with $m_{\text{RFE}}(\epsilon, \delta) \leq \mathcal{O}(d_{\text{lin}}^3 H^6 \iota^2/\epsilon^2)$, an approachability algorithm (Definition 3) with $m_{\text{APP}}(\epsilon, \delta) \leq \mathcal{O}(d_{\text{lin}}^3 H^6 \iota^2/\epsilon^2)$ and an algorithm for CMDP (Definition 4) with $m_{\text{CMDP}}(\epsilon, \delta) \leq \mathcal{O}(d_{\text{lin}}^3 H^6 \iota^3/\epsilon^2)$.*

The reward-free algorithm with stated sample complexity in Theorem 9 is the Algorithm 6 in Appendix D. it is a modified version of the reward-free algorithm introduced by Wang et al. (2020). Its sample complexity result is again obtained by adapting the results in Wang et al. (2020) for scalar-valued MDPs to the settings of VMDPs in this paper. The algorithms for approachability and CMDP is based on pluging in this reward-free algorithm into our meta algorithms, and the corresponding sample complexity results are obtained by applying Theorem 5 and our main result—Theorem 6.

Theorem 9 provides a new sample complexity result of $\tilde{\mathcal{O}}(d_{\text{lin}}^3 H^6/\epsilon^2)$ for the reward-free exploration (Definition 1) in the linear setting (Assumption 8). It further provides a new sample complexity result of $\tilde{\mathcal{O}}(d_{\text{lin}}^3 H^6/\epsilon^2)$ for both approachability (Definition 3) and CMDP (Definition 4) in the linear setting (Assumption 8). To best our knowledge, this is the first sample-efficient result for constrained RL problems with linear function approximation and general convex constraints.

# 6 VECTOR-VALUED MARKOV GAMES

## 6.1 MODEL AND PRELIMINARIES

Similar to Section 2, we consider an episodic *vector-valued Markov game* (VMG) specified by a tuple $\mathcal{G} = (\mathcal{S}, \mathcal{A}, \mathcal{B}, H, \mathbb{P}, \mathbf{r})$, where $\mathcal{A}$ and $\mathcal{B}$ are the action spaces for the min-player and max-player,

---

[2]This $H$ factor difference is due the Bernstein-type bonus used in Yu et al. (2021), which can not be adapted to the reward-free setting.

respectively. The $d$-dimensional return function $\mathbf{r}$ and the transition probabilities $\mathbb{P}$, now depend on the current state and the action of both players.

**Interaction protocol.**    In each episode, we start at a *fixed* initial state $s_1$. Then, at each step $h \in [H]$, both players observe the current state $s_h$, take their own actions $a_h \in \mathcal{A}$ and $b_h \in \mathcal{B}$ simultaneously, observe stochastic sample of the return vector $\mathbf{r}_h(s_h, a_h, b_h)$ along with their opponent's action, and it causes the environment to transit to $s_{h+1} \sim \mathbb{P}_h(\cdot \mid s_h, a_h, b_h)$. We assume that stochastic samples of the return function are also in $\mathcal{B}(1)$, almost surely.

**Policy and value function.**    A policy $\mu$ of the min-player is a collection of $H$ functions $\{\mu_h : \mathcal{S} \to \Delta(\mathcal{A})\}_{h=1}^H$. Similarly, a policy $\nu$ of the max-player is a collection of $H$ functions $\{\nu_h : \mathcal{S} \to \Delta(\mathcal{B})\}_{h=1}^H$. If the players are following $\mu$ and $\nu$, we have $a_h \sim \mu(\cdot|s)$ and $b_h \sim \nu(\cdot|s)$ at the $h^{\text{th}}$ step. We use $\mathbf{V}_h^{\mu,\nu} : \mathcal{S} \to \mathcal{B}(H)$ and $\mathbf{Q}_h^{\mu,\nu} : \mathcal{S} \times \mathcal{A} \times \mathcal{B} \to \mathcal{B}(H)$ to denote the value function and Q-value function at step $h$ under policies $\mu$ and $\nu$.

**Scalarized markov game and Nash equilibrium.**    For a VMG $\mathcal{G}$ and $\boldsymbol{\theta} \in \mathcal{B}(1)$, we define scalar-valued Markov game $\mathcal{G}_{\boldsymbol{\theta}} = (\mathcal{S}, \mathcal{A}, H, \mathbb{P}, r_{\boldsymbol{\theta}})$, where $r_{\boldsymbol{\theta}} = \{\langle \boldsymbol{\theta}, \mathbf{r}_h \rangle : \mathcal{S} \times \mathcal{A} \times \mathcal{B} \to [-1, 1]\}_{h=1}^H$. We use $V_h^{\mu,\nu}(\cdot; \boldsymbol{\theta})$ and $Q_h^{\mu,\nu}(\cdot, \cdot, \cdot; \boldsymbol{\theta})$ to denote value function and Q-value function of $\mathcal{G}_{\boldsymbol{\theta}}$, respectively. Note that we have $V_h^{\mu,\nu}(s; \boldsymbol{\theta}) = \langle \boldsymbol{\theta}, \mathbf{V}_h^{\mu,\nu}(s) \rangle$ and $Q_h^{\mu,\nu}(s, a, b; \boldsymbol{\theta}) = \langle \boldsymbol{\theta}, \mathbf{Q}_h^{\mu,\nu}(s, a, b) \rangle$.

For any policy of the min-player $\mu$, there exists a *best-response* policy $\nu_\dagger(\mu)$ of the max-player; i.e. $V_h^{\mu,\nu_\dagger(\mu)}(s; \boldsymbol{\theta}) = \max_\nu V_h^{\mu,\nu}(s; \boldsymbol{\theta})$ for all $(s, h) \in \mathcal{S} \times [H]$. We use $V^{\mu,\dagger}$ to denote $V^{\mu,\nu_\dagger(\mu)}$. Similarly, we can define $\mu_\dagger(\nu)$ and $V^{\dagger,\nu}$. We further know (Filar & Vrieze, 2012) that there exist policies $(\mu^\star, \nu^\star)$, known as *Nash equilibrium*, satisfying the following equation for all $(s, h) \in \mathcal{S} \times [H]$:

$$\min_\mu \max_\nu V_h^{\mu,\nu}(s; \boldsymbol{\theta}) = V_h^{\mu^\star,\dagger}(s; \boldsymbol{\theta}) = V_h^{\mu^\star,\nu^\star}(s; \boldsymbol{\theta}) = V_h^{\dagger,\nu^\star}(s; \boldsymbol{\theta}) = \max_\nu \min_\mu V^{\mu,\nu}(s; \boldsymbol{\theta})$$

In words, it means that no player can gain anything by changing her own policy. We abbreviate $V_h^{\mu^\star,\nu^\star}$ and $Q_h^{\mu^\star,\nu^\star}$ as $V_h^\star$ and $Q_h^\star$.

### 6.1.1    REWARD-FREE EXPLORATION (RFE) FOR VMGS

Similar to Section 2.1, we can define RFE algorithm for VMGs. Similarly, it consists of two phases. In the exploration phase the it explores the environment without guidance of return function. Later, in the planning phase, given any $\boldsymbol{\theta} \in \mathcal{B}(1)$, it requires to output near optimal Nash equilibrium for $\mathcal{G}_{\boldsymbol{\theta}}$.

**Definition 10** (RFE algorithm for VMGs). For any $\epsilon, \delta > 0$, after collecting $m_{\text{RFE}}(\epsilon, \delta)$ episodes during the exploration phase, with probability at least $1 - \delta$, the algorithm for all $\boldsymbol{\theta} \in \mathcal{B}(1)$, satisfies

$$V_1^{\mu_{\boldsymbol{\theta}},\dagger}(s_1; \boldsymbol{\theta}) - V_1^{\dagger,\nu_{\boldsymbol{\theta}}}(s_1; \boldsymbol{\theta}) = [V_1^{\mu_{\boldsymbol{\theta}},\dagger}(s_1; \boldsymbol{\theta}) - V_1^\star(s_1; \boldsymbol{\theta})] + [V_1^\star(s_1; \boldsymbol{\theta}) - V_1^{\dagger,\nu_{\boldsymbol{\theta}}}(s_1; \boldsymbol{\theta})] \le \epsilon$$

where $(\mu_{\boldsymbol{\theta}}, \nu_{\boldsymbol{\theta}})$ is the output of the planning phase for vector $\boldsymbol{\theta}$ as input. The function $m_{\text{RFE}}$ determines the *sample complexity* of the RFE algorithm.

### 6.1.2    BLACKWELL APPROACHABILITY FOR VMGS

We assume we are given a VMG $\mathcal{G}$ and a target set $\mathcal{C}$. The goal of the min-player is for the return vector to lie in the set $\mathcal{C}$ while max-player wants the opposite. For the two-player vector-valued games it can be easily shown that the minimax theorem does no longer hold (see Section 2.1 of Abernethy et al. 2011). Namely, if for every policy of the max-player we have a response such that the return is in the set, we cannot hope to find a single policy for the min-player so that for every policy of the max-player the return vector lie in the set. However, approaching the set on average is possible.

**Definition 11** (Blackwell approachability). We say the min-player is approaching the target $\mathcal{C}$ with rate $\alpha(T)$, if for arbitrary sequence of max-player polices $\nu^1, \ldots, \nu^T$, we have

$$\text{dist}(\tfrac{1}{T} \textstyle\sum_{t=1}^T \mathbf{V}_1^{\mu^t,\nu^t}(s_1), \mathcal{C}) \le \max_\nu \min_\mu \text{dist}(\mathbf{V}_1^{\mu,\nu}(s_1), \mathcal{C}) + \alpha(T).$$

## 6.2 META-ALGORITHM FOR VMGs

Similar to Section 3, we introduce our main algorithmic framework for VMGs bridging reward-free algorithm and Blackwell approachability in VMGs. The pseudo-code is displayed in Algorithm 2 and the theoretical guarantees are provided in Theorem 12. The proof can be found in Appendix E.

---

**Algorithm 2** Meta-algorithm for VMGs

---

1: **Input:** Reward-Free Algorithm RFE for VMG (as in Definition 10), Target Set $\mathcal{C}$
2: **Hyperparameters:** learning rate $\eta^t$
3: **Initialize:** run exploration phase of RFE for $K$ episodes
4: **Set:** $\boldsymbol{\theta}^1 \in \mathcal{B}(1)$
5: **for** $t = 1, 2, \ldots, T$ **do**
6:     Obtain near optimal Nash equilibrium for $\mathcal{G}_{\boldsymbol{\theta}^t}$:

$$(\mu^t, \omega^t) \leftarrow \text{output of planning phase of RFE for the vector } \boldsymbol{\theta}^t \text{ as input}$$

7:     Play $\mu^t$ for one episode:

$$\text{Play } \mu^t \text{ against max-player playing arbitrary policy } \nu^t \text{ for one episode}$$
$$\text{and let } \widehat{\mathbf{v}}^t \text{ be the sum of vectorial returns}$$

8:     Apply online gradient ascent update for utility function $u^t(\boldsymbol{\theta}) = \langle \boldsymbol{\theta}, \widehat{\mathbf{v}}^t \rangle - \max_{\mathbf{x} \in \mathcal{C}} \langle \boldsymbol{\theta}, \mathbf{x} \rangle$:

$$\boldsymbol{\theta}^{t+1} \leftarrow \Gamma_{\mathcal{B}(1)}[\boldsymbol{\theta}^t + \eta^t(\widehat{\mathbf{v}}^t - \text{argmax}_{\mathbf{x} \in \mathcal{C}} \langle \boldsymbol{\theta}^t, \mathbf{x} \rangle)]$$

    where $\Gamma_{\mathcal{B}(1)}$ is the projection into Euclidean unit ball

---

**Theorem 12.** *For any choice of RFE algorithm (Definition 10) and for any $\epsilon \in (0, H]$ and $\delta \in (0, 1]$, if we choose $K = m_{\text{RFE}}(\epsilon/2, \delta/2)$ and $\eta^t = \sqrt{1/H^2 t}$ ; then, with probability at least $1 - \delta$, the min-player in Algorithm 2, satisfies Definition 11 with rate $\alpha(T) = \mathcal{O}(\epsilon/2 + \sqrt{H^2 \iota/T})$ where $\iota = \log(d/\delta)$. Therefore to obtain $\epsilon$-optimality, the total sample complexity scales with $\mathcal{O}(m_{\text{RFE}}(\epsilon/2, \delta/2) + H^2 \iota/\epsilon^2)$.*

## 6.3 TABULAR VMGs

In this section, we consider tabular VMDPs; namely, we assume that $|\mathcal{S}| \leq S$, $|\mathcal{A}| \leq A$, and $|\mathcal{B}| \leq B$. Similar to Section 4, by utilizing VI-Zero (Liu et al., 2020) we can have the following theoretical guarantees. The algorithm and the proof can be found in Appendix E.

**Theorem 13.** *There exists a reward-free algorithm for tabular VMGs and a right choice of hyperparameters that satisfies Definition 10 with sample complexity $m_{\text{RFE}}(\epsilon, \delta) \leq \mathcal{O}(\min\{d, S\}H^4 SAB\iota/\epsilon^2 + H^3 S^2 AB\iota^2/\epsilon)$, where $\iota = \log[dSABH/(\epsilon\delta)]$.*

The theorem provides a new sample complexity result of $\tilde{\mathcal{O}}(\min\{d, S\}H^4 SAB\iota/\epsilon^2)$ for reward-free exploration in VMGs (Definition 10). It immediately follows from Theorem 13 and Theorem 12 that we can achieve total sample complexity of $\tilde{\mathcal{O}}(\min\{d, S\}H^4 SAB\iota/\epsilon^2)$ for Blackwell approachability in VMGs (Definition 11). Our rate for $\alpha(T)$ scales with $\tilde{\mathcal{O}}(\sqrt{\text{poly}(H)/T})$ while the results in Yu et al. (2021) has the rate of $\alpha(T)$ scaling with $\tilde{\mathcal{O}}(\sqrt{\text{poly}(H)\min\{d, S\}SA/T})$. However, we require initial phase of self-play for $K = \mathcal{O}(m_{\text{RFE}})$ episodes which is not needed by Yu et al. (2021).

## 7 CONCLUSION

This paper provides a meta algorithm that takes a reward-free RL solver, and convert it to an algorithm for solving constrained RL problems. Our framework enables the direct translation of any progress in reward-free RL to constrained RL setting. Utilizing existing reward-free solvers, our framework provides sharp sample complexity results for constrained RL in tabular setting (matching best existing results up to factor of horizon dependence), new results for the linear function approximation setting. Our framework further extends to tabular two-player vector-valued Markov games for solving Blackwell approachability problem.

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
