# OpenReview forum: "A Simple Reward-free Approach to Constrained Reinforcement Learning"
_ICLR.cc/2022/Conference — ICLR 2022 Submitted_

### Official Review · Reviewer_qB13 · 2021-10-22

**Correctness:** 4
**Technical Novelty And Significance:** 3
**Empirical Novelty And Significance:** 1
**Recommendation:** 5
**Confidence:** 3

**Main Review:**

I enjoyed reading this paper and I think that it has numerous merits:
- The paper is clear and very well-written.
- It studies an interesting topic for which optimal solutions are still not clear.
- The connection between reward-free RL and constrained RL is very clever and presented in an intuitive way.
- The resulting approach is indeed simple as the title suggests.
- The approach is general and gives sample complexity guarantees to any constrained RL formulation for which reward-free algorithms exist.
- I like the unification of constrained MDPs and approachability problems.

However, there are still quite a few things that prevent me from rating this paper as a clear accept:
- In general I think that the reward-free approach is celebrated by the authors a little too much without addressing its flaws. Here are a few things that I thought about (and maybe there are even more things to address):
     1. Sample complexity vs regret. The authors do mention that "due to reward-free nature of our approach unlike Yu et al. (2021), we can no longer provide regret guarantees", but I think that this is a major downside to the authors approach that needs to be addressed in details. Moreover, I am wondering whether the reward-free approach is simple simply because it is sub-optimal but optimal algorithms haven't been discovered yet. Can the authors please elaborate on regret?
     2. Lower bound. Are there any lower bounds for constrained MDPs? the authors do not mention them at all, and I think that they are crucial in order to put the presented algorithms and their guarantees in perspective. This is very related to the previous section, but again my guess is that the given guarantees have sub-optimal dependence in the number of states that may be removable with other approaches, but cannot be removed with the reward-free approach.
     3. The sub-optimality also comes into play in the $H$ dependence. The authors mention that "This H factor difference is due the Bernstein-type bonus used in Yu et al. (2021), which can not be adapted to the reward-free setting", and again this reveals the big downside of the reward-free approach which is probably sub-optimal as far as I understand.
     4. Computational complexity. The authors do not discuss computational complexity at all. How does the computational complexity of the proposed approach compare with other approaches? I am not sure what is the answer, but if it is worse then this is definitely another shortcoming of the presented approach. Anyway, it should absolutely be discussed in the paper.
- There is not a single proof sketch in this paper! In my opinion presenting sketches for theorems 5 and 6 are crucial in order to clearly convey the ideas of the paper, and it looks to me like the authors could easily find space for them (many models are presented in much detail, and I think that some of them, e.g., linear MDPs or MGs, could go to the appendix).
     1. I read the proof of theorem 5 in the appendix and I think that it presents a very nice idea that can and should be shown in 2-3 paragraphs in the main paper.
     2. Theorem 6 is the main contribution of the paper and an analysis overview must be provided in the main paper including the main techniques used by the authors.

**Summary Of The Paper:**

This paper proposes a meta-algorithm that takes an algorithm for reward-free RL and uses it to compute an (approximately) optimal policy for a constrained RL problem or an approachability problem. The proposed method keeps similar sample complexity guarantees to the ones provided by the reward-free algorithm, i.e., there is very small overhead.

**Summary Of The Review:**

The paper presents a simple approach to an interesting (and still open) problem, but in my opinion the mentioned flaws of the approach must be addressed in much detail before the paper is published.

---

> ### Author Response · Authors · 2021-11-21
> **Author Response**
>
> We would like to thank Reviewer qB13 for providing us with constructive feedback. We hope that our rebuttal will help clarify our work's novelty and contribution and convince the reviewer that our work warrants their acceptance.
>
> 1. First, we want to note that we can always convert sample complexity guarantees of $\tilde{\mathcal{O}}(\frac{1}{\epsilon^2})$ to regret guarantees using the explore-then-commit strategy resulting in a regret guarantee that scales with $\tilde{\mathcal{O}}(T^{2/3})$. However, the regret guarantee being sup-optimal is due to the reward-free nature of the algorithm and the fact that we don't have any reward information in the exploration phase.
> 2. There exist the lower bound of $\tilde{\Omega}(dH^3SA/\epsilon^2)$ (Section 6 in Yu et al. 2021). We'll make sure to include it in the final version.
> 3. Although we have this $H$ factor difference, our framework is more general, modular, and easily extended to settings beyond tabular (e.g., linear function approximation).
> 4. The computational complexity of our algorithm is in the same order as our sample complexity, i.e., [Computational Complexity] < $\mathcal{O}(d) \cdot $ [Sample Complexity]. We will make sure that we include the discussion regarding computational complexity in the final version.
> 5. Not having the proof sketch is mainly due to the space limit; we make sure to include it in the final version (since we have an extra page).

---

> > ### Comment · Reviewer_qB13 · 2021-11-30
> > **Keeping my score**
> >
> > I would like to thank the authors for the clarifications. However, I think that my original score is appropriate for this version of the paper.

---

### Official Review · Reviewer_ohJk · 2021-10-28

**Correctness:** 4
**Technical Novelty And Significance:** 2
**Empirical Novelty And Significance:** Not applicable
**Recommendation:** 5
**Confidence:** 3

**Main Review:**


While the paper is well executed and seems sound to me (I didn't have time to check all the proofs), I have doubts due to the limited technical novelty. I understand the proposed meta-structure allows to account for multiple problems, but the core idea is not very novel.
At a high level, the idea is that the considered problems can be formulated as a saddle point problem that can be solved with different incremental techniques. A standard approach is to use a dual approach where a best response algorithm is used to solve for the optimal policy and an incremental (gradient) algorithm is used to update the other player.
This approach was studied for example in (Miryoosefi et al., 2019) or (Efroni et al., 2020). In particular, the second approach used an estimate of the model to compute an optimistic policy (in the best response step), and they were able to prove regret guarantees. In this paper, the idea is to leverage reward-free techniques instead of estimating the model. Hence my doubts.
However, I acknowledge that, as far as I know, the proposed framework allows to derive results in new settings.

Furthermore, I think the result can be improved. In particular, the authors claim that Bernstein techniques cannot be used for RFE. However, [1] shows the opposite. Through the use of Bernstein technique, they were able to improve the sample complexity by a factor $H$ in RFE. So, could you please explain why it is not possible in this setting? I think this is quite a limitation of the current paper.


Minor comments:
- In section 3, you said that the minmax problem is convex in $\pi$. It should be clarified since it's not convex in $\pi$ but rather in the occupancy measure of a policy.
- First paragraph of Sec 3: ready introduce -> ready to introduce
- First paragraph of Sec 6.1.1: phase the it -> phase, it


[1] Menard et al. Fast active learning for pure exploration in reinforcement learning. ICML 2021



**Summary Of The Paper:**

The paper introduces a connection between reward-free and constrained MDPs. The interesting result in the paper is the fact that it is possible to use any reward-free method to solve constrained and approachability problems. In particular, they show that starting from an RFE method with sample guarantees, it is possible to obtain guarantees in these settings paying only a small cost. This meta-framework can be used to solve many different problems (e.g., tabular and linear problems).


**Summary Of The Review:**

I have concerns about novelty and contributions, and the fact the bounds may be improved.

---

> ### Author Response · Authors · 2021-11-21
> **Author Response**
>
> We would like to thank Reviewer ohJk for providing us with constructive feedback and also bringing our attention to [1]. We hope that our rebuttal will help clarify our work's novelty and contribution and convince the reviewer that our work warrants their acceptance.
>
> - We believe the idea that constrained RL can be reduced to vector-valued RFE is novel and has never been considered in the literature. Given their tight connections in sample complexity and the fact that our approach is clean and modular with the ability to easily extend to more challenging settings, it results in an important contribution to the community.
> - First, we note that the result in [1] has square dependence on $S$ while our results depend on $\text{min}\\{d, S\\}S$. It seems that square dependence in their approach is unavoidable, making it unsuitable for settings in which $ d \ll S$ that usually holds for CMDPs. Second, as we mentioned, the paper's main contribution is the seamless translation of results in RFE to Approachability and Constrained RL setting. The algorithm in [1] satisfies the definition of RFE for VMDPs in our paper (Definition 1); therefore, it automatically gives us the sample complexity guarantee of  $\tilde{\mathcal{O}}(H^3S^2A/\epsilon^2)$.
>
> [1] Menard et al. Fast active learning for pure exploration in reinforcement learning. ICML 2021

---

> > ### Comment · Reviewer_ohJk · 2021-11-30
> > **Thank you**
> >
> > Thank you for the answers. I'm willing to increase my score to 6, but I still believe the paper stands in a borderline position.

---

### Official Review · Reviewer_zcw7 · 2021-11-01

**Correctness:** 4
**Technical Novelty And Significance:** 2
**Empirical Novelty And Significance:** Not applicable
**Recommendation:** 6
**Confidence:** 3

**Main Review:**

## Strengths:

- Theorem 5 is a really interesting result. This result will allow the community to use advances in the approachability setting and to the constrained RL. This allows further advances in reward-free RL to directly translate in the approachability and constrained RL setting.
- The paper is well-written and generally clear.

## Weaknesses:
- **Constraints violation during learning?**: I’m unsure of the implication of the results in the Constrained-RL case. When working with the CMDP formulation, constraint violation during learning is a major motivation and it also brings the additional challenges of working with them (safe-exploration). I find it concerning that there is no mention of results regarding the constraint violation during the learning of the proposed algorithms or even the associated related work such as [1,2].  For instance, for tabular CMDPs and linear constraints, Liu et al [1] provide an approach that achieves $\mathcal{O)(1)}$ constraint violation in the general setting while still achieving regret bound of $\mathcal{\tilde{O}(H^3)}$. I think it is important to address the constraint violation in the exploration phase, especially in terms of CMDPs, and I wonder how do the authors approach fares in this regard.

- **Too much content?:** Unfortunately, the paper at times feels rushed and the reader is left out to figure the details to themselves. In particular, I felt that Section 3 is the core component but it is covered in not much detail.
This concern also follows the other parts of the work.  In the linear VMDPs section, applying Wang 2020 to the slightly modified setting gives us the new result for linear function approximation for constrained-RL. Although this result is new, I’m not sure if this is particularly useful. If it is, I fail to see how. Maybe the authors can motivate this further?

  The setting also changes significantly from MDPs to Markov games. I’m not sure if squeezing this component in the last two pages was the best approach. Moreover, the motivation for VMGs is missing from the introduction. I also don’t get why we don’t have a constrained-RL equivalent for the VMGs (Does the Blackwell approachability encompasses that component too?).  I’m not familiar with the literature from Markov games so my concerns with this section should be taken with a grain of salt.

- **Empirical experiments:** Although, I understand that the main contribution of the work is theoretical, however empirical experiments in the constrained-RL setting can shed some light on how the algorithm performs in practice wrt the behaviour during the learning.

**Summary Of The Paper:**

The paper applies the techniques from reward-free RL literature to the constrained RL setting. They propose a meta-algorithm that takes a reward-free RL solver and uses it to solve the approachability and constrained-RL problems with convex constraints. The proposed approach comes with an overhead factor that is logarithmic in the number of samples.   They apply this approach to provide sharp analysis on three different settings: tabular VMDPs, linear VMDPs and two-player VMGs.

**Summary Of The Review:**

I think there are some really interesting results in this work, however, there seems to be a conflict in the motivation (constrained-RL problems) and methodology (reward-free RL solutions). On a high level, the exploration phase in the reward-free RL seems orthogonal to the safe exploration problem in Constrained-RL. If exploration is not an issue, then the motivation behind the tabular and linear VMDPs is not clear (as one can use other CMDP solvers).  I wonder applying the methods from the reward-free literature to the constrained-RL setting gives us any meaningful results. I think in the current form, it is not clear what the benefits of such an approach are. Hence, I’m voting for a weak reject. I’m happy to revise the score if the authors can further motivate the benefits of applying RL-free methods to the constrained-RL setting.

---

> ### Author Response · Authors · 2021-11-21
> **Author Response**
>
> We would like to thank Reviewer zcw7 for spending time reading our paper and providing us feedback. We hope that our rebuttal will help clarify our work's novelty and contribution, and convince the reviewer that our work warrants their acceptance.
>
> First, we want to note that we can always convert sample complexity guarantees of $\tilde{\mathcal{O}}(\frac{1}{\epsilon^2})$ to regret guarantees using the explore-then-commit strategy resulting in a regret guarantee that scales with $\tilde{\mathcal{O}}(T^{2/3})$. Second, the papers mentioned by the reviewer (and other similar results) are only limited to tabular settings with only linear constraints, while our framework can handle general convex constraints and linear function approximation settings. Third, we want to mention that our framework allows us to change the constraint set to a new one with a negligible overhead which is not feasible in the prior work (the constraint set should be fixed before learning starts).  Our main contribution is this modular and general framework that seamlessly translates advances in RFE literature to sample complexity guarantees for Approachability and Constrained Reinforcement Learning. This modularity comes at some cost, .e.g, achieving regret $\tilde{\mathcal{O}}(T^{2/3})$ instead of $\tilde{\mathcal{O}}(\sqrt{T})$. Although we agree that some areas are still open for future research, we strongly believe that the merits of the proposed framework outweigh the minor flaws.
>
> We believe that contribution of our paper is purely theoretical, and our framework is the first algorithm to achieve efficient sample complexity guarantees for Approachability and Constrained RL in a linear function approximation setting (linear VMDPs).
>
> Regarding the CMDP equivalent in Markov games: to the best of our knowledge, there isn't any existing definition of constrained RL in the Markov game setting. In this setting, It's not clear if the opponent should compete with the player in both constraint and utility or only one of them.

---

### Decision · Program_Chairs · 2022-01-20

**Decision:**

Reject

**Comment:**

The paper presents a general solution method for constrained RL problems using reward-free exploration. While the reviewers found this reduction interesting in general, they had concerns about the price of this reduction in general (such as the increased regret or for suboptimal dependence of the bounds on some problem parameters), which is to be paid in exchange for the simplicity and flexibility of the proposed approach. This, coupled with the limited technical novelty used in the derivations, made all reviewers think that this is a borderline paper, and I also agree with this assessment. The paper could benefit a lot from presenting more evidence of the benefits of their approach (either theoretically or empirically). Based on the above, unfortunately, I am not able to recommend acceptance at this point.